# *hkb* is required for *DIP-*α expression and target recognition in the *Drosophila* neuromuscular circuit
Yupu Wang [1,2,5,6,7] ✉, Rio J. Salazar[1,2,3,6], Luciano T. Simonetta[1,2], Violet Sorrentino [1,2], Terrence J. Gatton[4], Bill Wu[4], Christopher G. Vecsey[4] & Robert A. Carrillo [1,2,3,7] ✉

Our nervous system contains billions of neurons that form precise connections with each other through interactions between cell surface proteins. In *Drosophila*, the Dpr and DIP immunoglobulin protein subfamilies form homophilic or heterophilic interactions to instruct synaptic connectivity, synaptic growth, and cell survival. However, the upstream regulatory mechanisms of Dprs and DIPs are not clear. On the other hand, while transcription factors have been implicated in target recognition, their downstream cell surface proteins remain mostly unknown. We conduct an F1 dominant modifier genetic screen to identify regulators of Dprs and DIPs. We identify *huckebein* (*hkb*), a transcription factor previously implicated in target recognition of the dorsal Is motor neuron. We show that *hkb* genetically interacts with *DIP-*α and loss of *hkb* leads to complete removal of *DIP-*α expression specifically in dorsal Is motor neurons. We then confirm that this specificity is through the dorsal Is motor neuron specific transcription factor, *even-skipped* (*eve*), which acts downstream of *hkb*. Analysis of the genetic interaction between *hkb* and *eve* reveals that they act in the same pathway to regulate dorsal Is motor neuron connectivity. Our study provides insight into the transcriptional regulation of *DIP-*α and suggests that distinct regulatory mechanisms exist for the same CSP in different neurons.

The way animals perceive and respond to the environment relies on precise and robust neuronal connections. During development, each neuron must identify the correct synaptic partners among thousands of potential targets. A prevalent model for instructing synaptic recognition, repulsion, and self-avoidance is through the interaction between unique cell surface proteins (CSPs). A major subset of CSPs belong to the immunoglobulin superfamily (IgSF), which play important roles in synaptic development and maintenance in both vertebrates and invertebrates. In the well-studied vertebrate retina, retinal ganglion cells require Dscams and Sidekicks (Sdks) 1 and 2 to avoid self-synapses and form stereotyped connections, respectively[1–3]. In hard-wired invertebrate nervous systems, such as *C. elegans*, the heterophilic interaction between two IgSF proteins, Syg1 and Syg2, is required for HSNL motor neuron (MN) synapse formation[4,5]. In the *Drosophila* mushroom body, neurons utilize different isoforms of Dscam1 to discriminate self/non-

self [6–8]. Several IgSF CSPs have also been implicated in synaptic connectivity in the *Drosophila* larval neuromuscular system where two type-Is motor neurons (Is MNs) and ~29 type-Ib motor neurons (Ib MNs) form stereotyped connections with 30 muscles in each hemisegment[9–11]. For example, the immunoglobulin proteins Fasciclin 2[12,13] and Fasciclin 3[14,15] are required for specific larval MNs to recognize their muscle targets.

Recent biochemical studies revealed two *Drosophila* immunoglobulin protein subfamilies, the Defective proboscis response proteins (Dprs, 21 members) and Dpr-interacting proteins (DIPs, 11 members) families, that form homophilic or heterophilic interactions to instruct synaptic connectivity, synaptic growth, and cell survival[16–29]. For example, the well-studied Dpr10-DIP-α interaction is necessary for innervation of the dorsal Is MN on muscle 4 (m4) as loss of either *dpr10* or *DIP-*α leads to complete loss of m4-Is innervation[24]. In addition, loss of *dpr10* or *DIP-*α in the optic lobe

[1]Department of Molecular Genetics and Cellular Biology, University of Chicago, Chicago, IL 60637, USA. [2]Neuroscience Institute, University of Chicago, Chicago, IL 60637, USA. [3]Program in Cell and Molecular Biology, University of Chicago, Chicago, IL 60637, USA. [4]Neuroscience Program, Skidmore College, 815 N. Broadway, Saratoga Springs, NY 12866, USA. [5]Present address: Howard Hughes Medical Institute, Janelia Research Campus, Ashburn, VA 20147, USA. [6]These authors contributed equally: Yupu Wang, Rio J. Salazar. [7]These authors jointly supervised this work: Yupu Wang, Robert A. Carrillo. ✉e-mail: wangy9@hhmi.org; robertcarrillo@uchicago.edu

causes significant mistargeting and cell death of Dm12 medulla neurons, suggesting multifaceted roles for Dpr10-DIP-α interactions[19,25]. Similarly, the recognition between yellow R7 photoreceptors (yR7) and yellow Dm8 neurons (yDm8) relies on the complementary expression of Dpr11 and DIP-γ, respectively, and the lack of either *dpr11* or *DIP-γ* leads to the failure of yR7 and yDm8 to recognize each other and subsequent yDm8 cell death[16,30,31]. Although extensive studies have uncovered roles for Dpr-DIP interactions, the regulation and downstream mechanisms are severely understudied.

Transcription factors (TFs) are the fate determinants of all cell types, and in neurons, they are master regulators of synaptic wiring by determining the expression of many factors including CSPs. In the fly olfactory system, cell type-specific expression of the TF Acj6 controls expression of a cell-surface code that instructs neurons to identify correct synaptic partners[32,33]. In the visual system, the homeodomain TF, Brain-specific homeobox (Bsh), directly binds to the *DIP-β* locus and other L4 identity genes to specify L4 neuronal fate[29]. In addition, stochastic expression of Spineless (ss) determines the yR7 fate and controls the expression of *dpr11*, which is required for synaptic connectivity with yellow Dm8s[30]. Similarly, during embryonic development, several key TFs specify the neuroblast (NB) lineages, including *huckebein* (*hkb*)[34–37], which is detected in 8 NB lineages and is required for the expression of the cell fate marker, *even-skipped* (*eve*), in NB4-2[38–40]. In *hkb* mutant embryos, RP2 MNs (also known as the dorsal Is MNs) derived from NB4-2 show severe wiring defects as they do not reach the correct muscles, suggesting that *hkb* controls specific CSPs for synaptic recognition in dorsal Is MNs[39]. However, unlike Acj6, Bsh, and Ss, the CSP(s) downstream of Hkb that control synaptic recognition are not known.

In this study, we sought to identify genes involved in connectivity by developing a sensitized genetic background with known wiring CSPs that

could be modified. Homozygous loss of *dpr10* and *DIP-α* led to complete loss of innervation of m4 by the dorsal Is MNs but *DIP-α/+;dpr10/+* trans-heterozygous larvae showed a 50% reduction of m4-Is innervation frequency. This trans-heterozygous background was used for an F1 deficiency screen to identify dominant enhancers or suppressors of Dpr10/DIP-α-mediated connectivity. We screened deficiency lines from the Bloomington Deficiency Kit covering the right arm of the third chromosome[41,42], and within one interacting line, we identified *hkb* as a genetic regulator of *DIP-α*. *DIP-α* is expressed in both dorsal and ventral Is MNs, but interestingly, we found that *hkb* is only necessary for *DIP-α* expression and MN-muscle recognition in the dorsal Is MN, suggesting distinct regulatory mechanisms for *DIP-α* in dorsal and ventral Is MNs. Next, we showed that *hkb* functions through the dorsal Is MN specific TF, *eve*, as *DIP-α* expression and dorsal Is MN innervation are also disrupted in *eve* mutants. Genetic interaction tests between *hkb* and *eve* further confirmed that they act in the same pathway. In summary, our study reveals that Hkb acts through Eve to control *DIP-α* gene expression to regulate MN-muscle connectivity, bridging the gap between upstream TFs and downstream CSPs. Moreover, our study suggests that distinct regulatory mechanisms exist for the same CSP in different neurons.

## Results
### Genetic screen identifies *hkb*, a genetic interactor of Dpr10-DIP-α pathway
The *Drosophila* larval neuromuscular system provides an ideal model to study genetic programs that instruct synaptic recognition due to the ease of genetic manipulation and the stereotyped connectivity patterns. Each larval body wall hemisegment is innervated by one ventral and one dorsal Is MN that connect to the ventral or dorsal muscle groups, respectively, in a stereotyped manner (Fig. 1a). In a previous study, we showed that among all

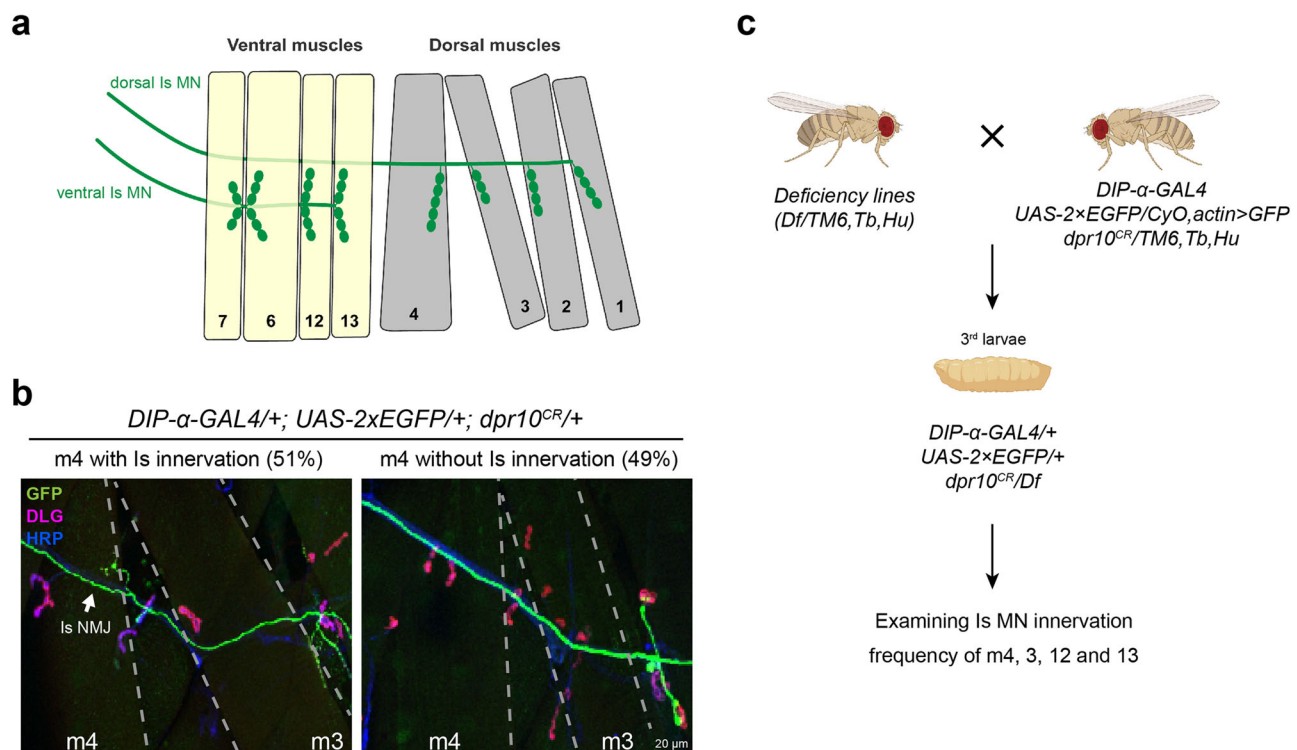

**Fig. 1 | Establishing a sensitized background for a deficiency screen. a** Cartoon depicting the innervation pattern of dorsal Is MN and ventral Is MN. **b** Representative images of muscle 4 with Is innervation or without Is innervation, in trans-heterozygotes of *DIP-α* (the *DIP-α-GAL4* is also a null allele) and *dpr10*. 51% of m4s are innervated by the dorsal Is MN. GFP (green), DLG (magenta) and HRP (blue) are shown in the images. Arrow pointing to the Is NMJ. **c** Workflow of

the deficiency screen. Male flies carrying the deficiency chromosome were crossed to females with *DIP-α* and *dpr10* mutations. Female third instar larvae were selected against the second and third chromosome balancers (*CyO,actin > GFP* and *TM6,Tb,Hu*). Triple-heterozygous larvae were dissected and Is innervation frequency on m4, 3, 12 and 13 were scored. Cartoon is created with BioRender.com.

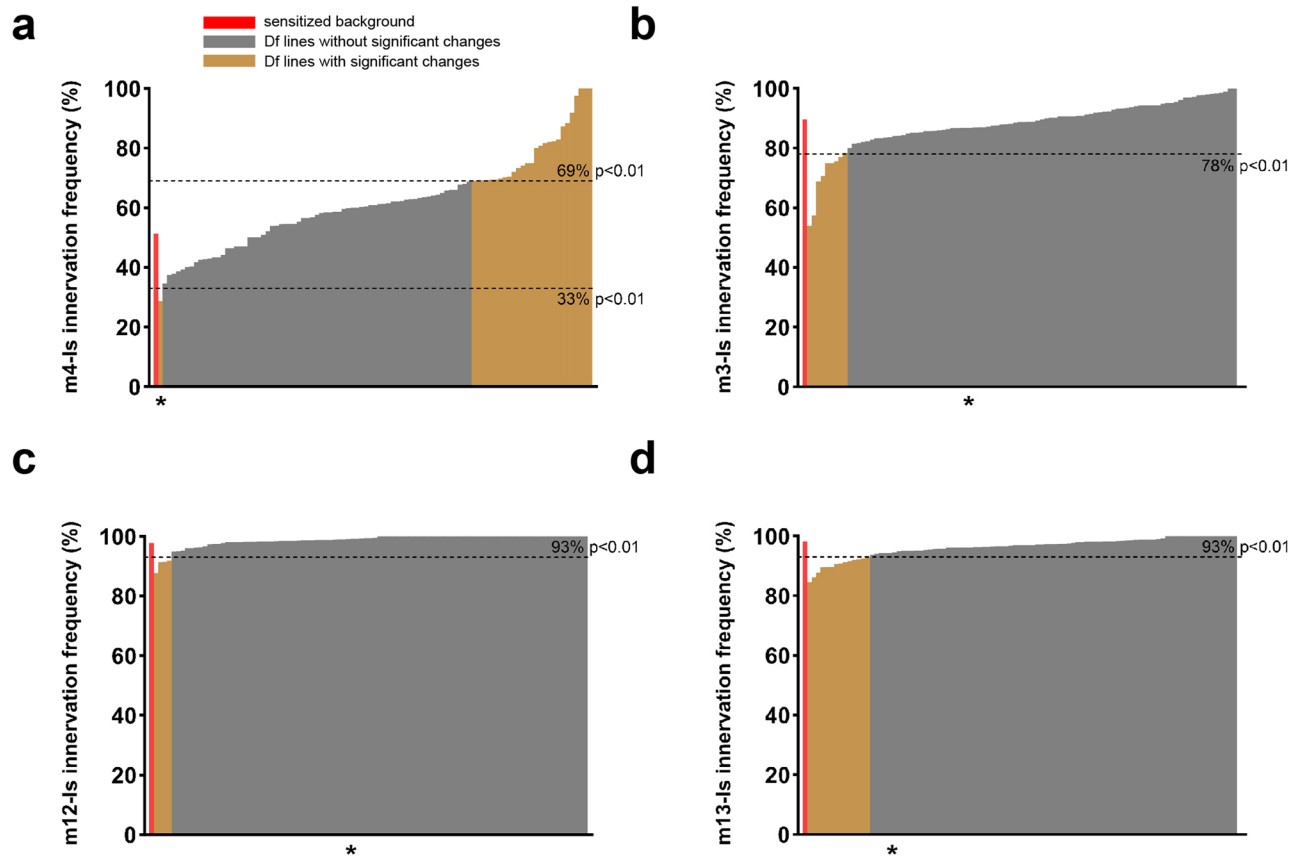

**Fig. 2 | Deficiency screen revealed candidate regions that cover genetic interactors of *DIP-α* or *dpr10*.** Is innervation frequency on (**a**) m4, (**b**) m3, (**c**) m12 and (**d**) m13. The red column indicates the control innervation frequency from the sensitized background (trans-heterozygotes of *DIP-α* and *dpr10*). Gray columns are non-significant from control whereas the yellow columns are the deficiency lines that show significantly different innervation frequencies compared to control. The cut-off *p* values are indicated by dashed lines. Asterisk indicates ED5100.

MNs, *DIP-α* is expressed exclusively in these two Is MNs, and its interacting partner, Dpr10, is expressed in a subset of muscles[10,24]. The interaction between Dpr10 and DIP-α is required for the recognition between dorsal Is MNs and several dorsal muscles. Specifically, loss of either *dpr10* or *DIP-α* leads to complete loss of dorsal Is MN innervation on m4, suggesting that Dpr10-DIP-α interaction is absolutely required for m4-Is innervation. This easily scorable phenotype prompted us to ask what other genes are involved in this Dpr10-DIP-α-dependent synaptic recognition. Because loss of either CSP results in complete loss of m4-Is connectivity and single heterozygotes have either no or very mild phenotypes (see below), we created a sensitized genetic background in which one copy of *dpr10* and *DIP-α* are removed. In this background, m4-Is innervation frequency is reduced to ~51% (Fig. 1b), compared to a 90% m4-Is innervation frequency in wild type animals[24]. We chose a *dpr10* CRISPR (*dpr10^CR*) allele and a GAL4 insertion allele of *DIP-α* (*DIP-α-GAL4*) derived from a MiMIC line, which disrupts endogenous *DIP-α* transcription and translation. Together with a *UAS-2xEGFP* construct, this *DIP-α-GAL4* allele aids identification of Is MN axons and neuromuscular junctions (NMJs) on different muscles since *DIP-α* is exclusively expressed in Is MNs (Fig. 1b). The reduced m4-Is innervation frequency in the sensitized background allowed us to screen for genetic interactors of the Dpr10-DIP-α pathway by introducing other mutations – if a mutation exacerbates or suppresses the decreased Is MN innervation on m4, we hypothesize that the gene may be part of the Dpr10-DIP-α pathway.

To improve the throughput, we utilized the Bloomington Deficiency (Df) kit and conducted an F1 dominant modifier screen (Fig. 1c). We screened 105 Df lines that cover the entire *Drosophila* chromosome 3R. Each Df line was combined into the sensitized background to create triple heterozygotes and the m4-Is innervation frequency was quantified (Fig. 2a). In addition, we also quantified the innervation frequency of Is MNs on other muscles, including dorsal muscle 3 (m3) and ventral muscle 12 (m12) and muscle 13 (m13) (Fig. 2b–d). Although the Is innervation frequencies on these muscles were not significantly decreased in the sensitized background, we hypothesized that genetic interactors or redundant molecules of the Dpr10-DIP-α pathway may be uncovered. Compared to the sensitized background (red columns), we identified several Df lines that significantly increased or decreased Is innervation frequency (yellow columns) (Fig. 2). The ED5100 Df line reduced m4-Is innervation frequency most significantly ($p < 0.01$, Chi-square test), but did not affect Is innervation frequency on m12, m13 and m3 ($p > 0.05$, Chi-square test), suggesting that it covers a gene(s) that may positively regulate the Dpr10-DIP-α pathway in the dorsal Is MNs for m4-Is recognition.

ED5100 is a 900 kb deletion that spans several genes and long non-coding RNAs. To narrow down the genomic region that covers our gene(s) of interest, we conducted a sub-screen using additional Df lines, ED5142 and ED5046, which partially overlap with the deletion in ED5100 (Fig. 3a). We observed a similar decrease of m4-Is innervation frequency when the sensitized line was crossed to ED5046, but not to ED5142 (Fig. 3b), suggesting that our gene(s) of interest is located in the region of ED5046 that overlaps with ED5100 but not ED5142, from 4197 kb to 4453 kb (Fig. 3a). This candidate region covers 58 genes including protein coding genes and non-coding RNAs. We further assayed 8 candidate genes with known or putative neuronal functions and an available mutant stock, including *auxilin*, *abstrakt*, *complexin*, *vps24*, *hkb*, *contactin*, *tube* and *lost*. We screened each candidate by combining a heterozygous mutant allele into our sensitized background and examined m4-Is innervation frequency. Of these candidates, we found that heterozygous loss of *hkb* (*hkb²/+*) exacerbated the m4-Is innervation defect when combined with the sensitized background (Fig. 3c), suggesting that *hkb* genetically interacts with the Dpr10-DIP-α pathway.

**Fig. 3 | A sub-screen identified *huckebein* (*hkb*) as a genetic interactor. a** Cartoon depicting the deleted regions within deficiency lines ED5100, ED5142, and ED5046. **b** Quantification shows a significant reduction of m4-Is innervation when combining ED5046, but not ED5142, with the sensitized background, suggesting the shared region between the original deficiency line (ED5100) and ED5046 covers the candidate gene(s). N (NMJs) = 177, 196, 155 and 155. *p* values are indicated. **c** Quantification of sub-screen of individual genes from candidate region shown in (**a**). Alleles used to create triple-heterozygotes are, $aux^{D128}$, $abs^{00620}$, $cpx^{MI00784}$, $vps24^{EY04708}$, $hkb^2$, $cont^{G5080}$, $tub^2$, $lost^{EY11645}$. Note that *huckebein* (*hkb^2*) further reduced m4-Is innervation frequency. N (NMJs) = 177, 138, 62, 58, 29, 135, 39, 77, 78 and 57. *p* values are indicated.

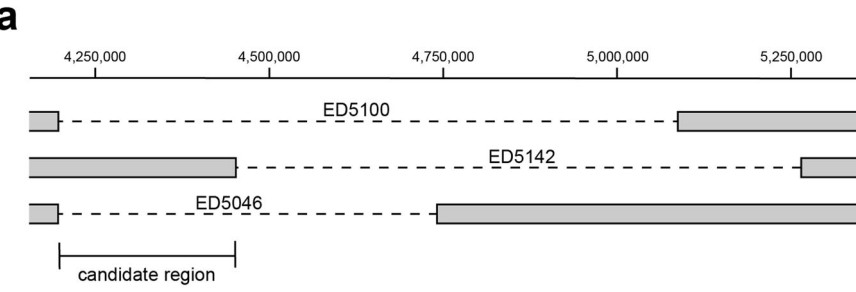

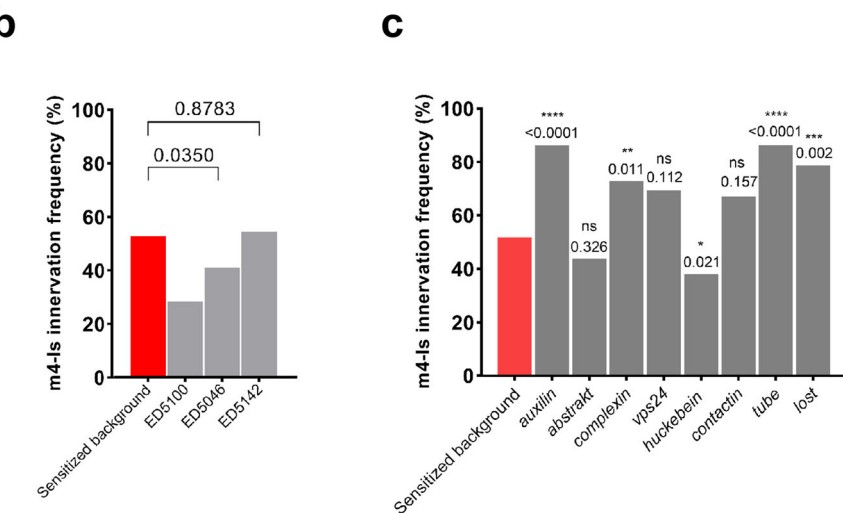

**Fig. 4 | *hkb* genetically interacts with *DIP-α*, but not *dpr10*. a** Genetic interaction assay between *hkb* and *dpr10*. Single heterozygotes of *dpr10* or *hkb* did not have altered m4-Is innervation, and neither did the trans-heterozygotes. N (NMJs) = 93, 154, 83, 125, 102 and 149. *p* values are indicated. **b** Genetic interaction assay between *hkb* and *DIP-α*. Single heterozygotes of *DIP-α* or *hkb* had slightly decreased m4-Is innervation frequency, while the trans-heterozygotes showed a further reduction, suggesting that *hkb* genetically interacts with *DIP-α*. N (NMJs) = 114, 105, 106, 103, 99 and 103. *p* values are indicated.

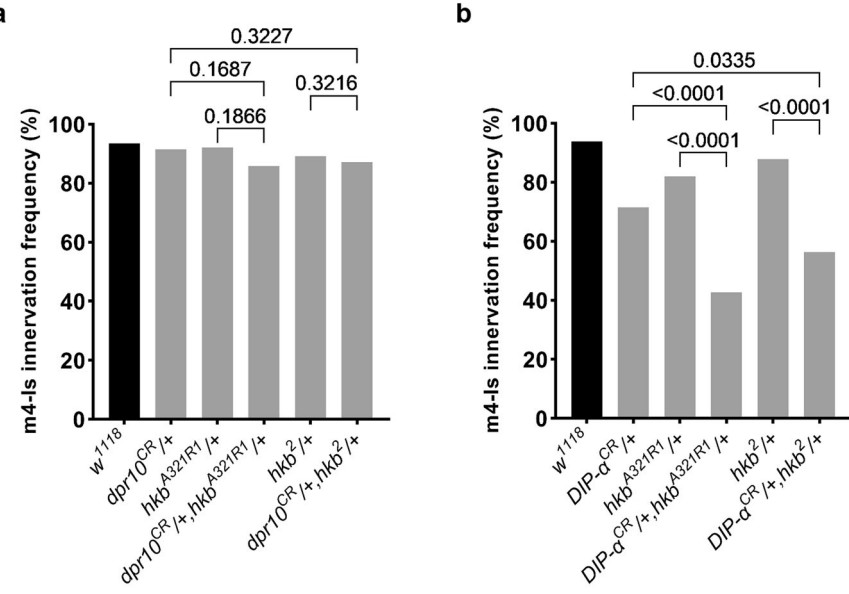

## *hkb* genetically interacts with *DIP-α*, but not *dpr10*

Our sensitized background is heterozygous for both *dpr10* and *DIP-α*. Therefore, *hkb* may genetically interact with either or both CSPs. Here, we examined genetic interaction between *hkb* and *dpr10* or *DIP-α* in trans-heterozygous animals. We combined two different *hkb* mutant alleles (*hkb^2* or *hkb^{A321R1}*) with a heterozygous *dpr10* mutant or *DIP-α* mutant and examined the m4-Is innervation frequency. In this and the following experiments, we used a *DIP-α* CRISPR (*DIP-α^{CR}*) allele since we will

primarily focus on m4-Is innervation and no longer need to identify Is NMJs on different muscles using *DIP-α-GAL4*.

In wild type animals, m4s are innervated about 90% of the time by the dorsal Is MN, and single heterozygotes of *dpr10* or *hkb* did not significantly decrease this innervation frequency (Fig. 4a). We then examined trans-heterozygotes of *dpr10* and *hkb* and found that the m4-Is innervation frequency was not significantly changed compared to single heterozygotes (Fig. 4a), suggesting that *hkb* is not a genetic interactor for *dpr10*. In contrast,

heterozygous loss of *DIP-α* reduced m4-Is innervation frequency to 71% (Fig. 4b), but the trans-heterozygotes of *DIP-α* and *hkb* further reduced the m4-Is innervation frequency to about 50%. Comparing the trans-heterozygous data with single heterozygotes (Fig. 4b) suggests that *hkb* and *DIP-α* are in the same genetic pathway. Overall, these data suggest that *hkb* genetically interacts with *DIP-α* but not *dpr10*.

### *hkb* controls *DIP-*α expression in the dorsal Is MNs

Next, we asked how *hkb* genetically interacts with *DIP-α*. *DIP-α* is expressed in both dorsal and ventral Is MNs but not in muscles, and interestingly, prior studies found that *hkb* is expressed in NB4-2, which produces the dorsal Is MN[38,39]. Therefore, we wondered if the TF *hkb* is required for *DIP-α* expression in dorsal Is MNs. To visualize *DIP-α* expression, we used an endogenously tagged *DIP-α-EGFP* allele[18]. In wild type animals, *DIP-α* is highly expressed in both dorsal and ventral Is MNs by stage 16 (Fig. 5a, b, b'). However, in *hkb²* mutant embryos, expression of *DIP-α* in dorsal Is MNs was completely lost (Fig. 5c). We confirmed loss of *DIP-α* in heteroallelic *hkb* mutant embryos (*hkb²/hkb^{A321R1}*) (Fig. 5d). Notably, the dorsal Is MN marker, *eve*, was also lost in *hkb* mutants as *hkb* is required for *eve* expression (Fig. 5c, d)[39]. These findings could be explained by the loss of the dorsal Is MNs; however, prior studies confirmed that dorsal Is MNs remain in *hkb* mutants even though they are *eve* negative[39,43]. Therefore, the lack of *DIP-α-EGFP* is not due to missing Is MNs, but to the loss of *hkb*. In addition, *DIP-α-EGFP* was not affected in the ventral Is MNs (Fig. 5c', d') or in other *DIP-α-EGFP* positive neurons (arrowheads in Fig. 5), suggesting that *hkb* only controls *DIP-α* expression in the dorsal Is MNs. This result is consistent with the unchanged ventral Is MN innervation frequency for ED5100 in our genetic screen. Taken together, these data indicate that different mechanisms regulate the same CSP in different neurons, and that *hkb* is required for *DIP-α* expression specifically in the dorsal Is MNs.

### *hkb* functions through *eve* to regulate *DIP-*α and dorsal Is MN innervation

As a TF, *hkb* may directly instruct *DIP-α* expression, or alternatively, function through other intermediate TFs. Prior studies reported that *hkb* is expressed early in the NB4-2 lineage, which gives rise to dorsal Is MNs, and turned off by stage 12, before synaptic recognition occurs in the neuro-muscular system[39]. However, a recent study found that *hkb* continues to be expressed in dorsal Is MNs during larval development[44]. Therefore, we decided to differentiate between the direct or indirect models of regulation. In NB4-2, a well-studied role of *hkb* is to trigger expression of the fate determinant TF, *eve*. Loss of *hkb* completely abolished *eve* expression in dorsal Is MNs (Fig. 5) (ref. 39). Thus, we wondered if *hkb* functions through *eve* to regulate *DIP-α* expression. Utilizing the *DIP-α-EGFP* allele and a conditional *eve* knock-out (*eve^{ΔRN2}*) which only lacks *eve* in dorsal Is MNs and siblings[43], we observed that *DIP-α* expression was lost in the dorsal Is MN, and the expression in ventral Is MNs was not affected, in both embryos and 1st instar larvae (Fig. 6a–e). Consequently, *eve* mutants lacked m4-Is innervation by the dorsal Is NMJs, whereas innervation by the ventral Is MNs was not affected (Fig. 6f–k). To further investigate the role of *eve* in synaptic recognition, we created *eve* and *DIP-α* trans-heterozygotes and examined m4-Is innervation frequency. Compared to single heterozygotes of *eve* or *DIP-α*, trans-heterozygotes significantly reduced m4-Is innervation frequency to about 60%, confirming that *eve* regulates m4-Is innervation by driving *DIP-α* expression in the dorsal Is MNs (Fig. 7a). Finally, we examined the trans-heterozygotes of *hkb* and *eve* and found that partial loss of both *hkb* and *eve* reduced m4-Is innervation frequency to 75%, suggesting that *hkb* and *eve* indeed act in the same pathway to control synaptic recognition of the dorsal Is MNs (Fig. 7b). Taken together, our results reveal a transcriptional cascade that regulates expression of wiring CSPs to guide MN-muscle recognition.

### Discussion

Synaptic recognition requires the interaction of CSPs, highlighting the critical role of regulatory programs to instruct the expression of CSPs in specific cells. Most studies have focused on the roles of TFs and CSPs independently, but less is known about the CSPs downstream of specific TFs in synaptic target recognition. For example, *hkb* and *eve* were implicated in pathfinding and target recognition of the dorsal Is MNs, but the molecules acting downstream of *hkb* and *eve* were unknown. On the other hand, the well-studied Dpr10-DIP-α interaction was found to guide the recognition between the dorsal Is MN and m4, but the regulatory mechanisms controlling the expression of these CSPs were not known[24]. To identify additional components in the Dpr10-DIP-α pathway, including transcriptional regulators, we conducted a dominant modifier genetic screen and found that *hkb* exacerbates the decrease of m4-Is innervation frequency when introduced into a *DIP-α* and *dpr10* trans-heterozygous background. Notably, our genetic screen identified many Df lines and candidate genes that rescue the m4-Is innervation frequency. This unexpected finding suggests that genes in these Df lines may be repulsive cues, repressors for synaptic growth, or are involved in the pruning process. However, all of these candidates may not act in the Dpr10-DIP-α pathway. For example, the triple heterozygote of *complexin* (*cpx*), *DIP-α*, and *dpr10* shows a wild type m4-Is innervation frequency (Fig. 3c), yet *cpx* mutants do not show connectivity phenotypes. Instead, loss of *cpx* revealed an increase of NMJ size and elevated activity[45,46]. The rescue in innervation observed in the triple heterozygote may be due to the perturbed synaptic activity which has been implicated in synaptic pruning[47,48]. A smaller subset of Df lines and candidate genes exacerbated the loss of m4-Is connectivity, and we chose to focus on these dominant modifiers. Specifically, we showed that *hkb* is required for *DIP-α* expression in dorsal Is MNs. Further examination revealed that *hkb* functions through *eve* to regulate *DIP-α* expression and dorsal Is MN innervation of m4, revealing a pathway linking TFs to specific CSPs and circuit assembly.

Interestingly, *hkb* is a gap gene originally implicated in embryonic development. In the early embryo, three pattern organizing centers, the anterior, the posterior, and the terminal, establish the anterior-posterior body plan by spatiotemporally regulating the expression of gap genes[49]. In the terminal control center, *torso* controls the expression of terminal gap genes including *hkb*[34]. *hkb* is expressed in the terminal cap during embryonic stages 5-6 and functions as a negative regulator to suppress gene expression in the terminal band, such as *odd-paired* (*opa*), *Dichaete* (*D*), and *caudal* (*cad*)[50]. In later embryonic stages, Hkb is expressed in a subset of NBs where it is required for glial development[51], serotonergic neuron differentiation[52,53], and for Eve expression to control motor axon pathfinding[39]. However, the molecules downstream of Hkb and Eve for axon pathfinding are not known, and additionally, the role of Hkb after the motor axon pathfinding stage has not been examined likely due to the lethality of null mutant embryos. In this study, we identified *hkb* as a *DIP-α* genetic interactor, and utilizing trans-heterozygous *hkb* hypomorph animals, we found that *hkb* is required for *DIP-α* expression to instruct innervation by the dorsal Is MN. Notably, our data suggests that *hkb* indirectly regulates *DIP-α* expression through the cell fate determinant TF, Eve. However, in a previous ChIP-Seq study profiling Eve target genes, many CSPs required for synaptic development were found, but *DIP-α* was not identified[54]. This could be due to *DIP-α* only being expressed in a small subset of Eve-expressing cells, or more likely, because Eve also indirectly regulates *DIP-α* expression since it mostly functions as a transcriptional repressor[43]. Nevertheless, our genetic analyses focused on a single cell type and revealed regulatory relationships that would be obscured in sequencing-based profiling.

Excitatory MNs in *Drosophila* larvae are classified into type-Ib and type-Is MNs due to their terminal bouton size and innervation patterns. Notably, *DIP-α* is selectively expressed in the dorsal and ventral Is MNs but is absent in Ib MNs. We therefore initially hypothesized that a common regulatory program may be responsible for the expression of *DIP-α* in both Is MNs but absent in Ib MNs. However, we found that *hkb* and *eve* regulate *DIP-α* expression specifically in dorsal Is MNs. *hkb* and *eve* are not expressed in ventral Is MNs (derived from NB3-1)[38], indicating that distinct

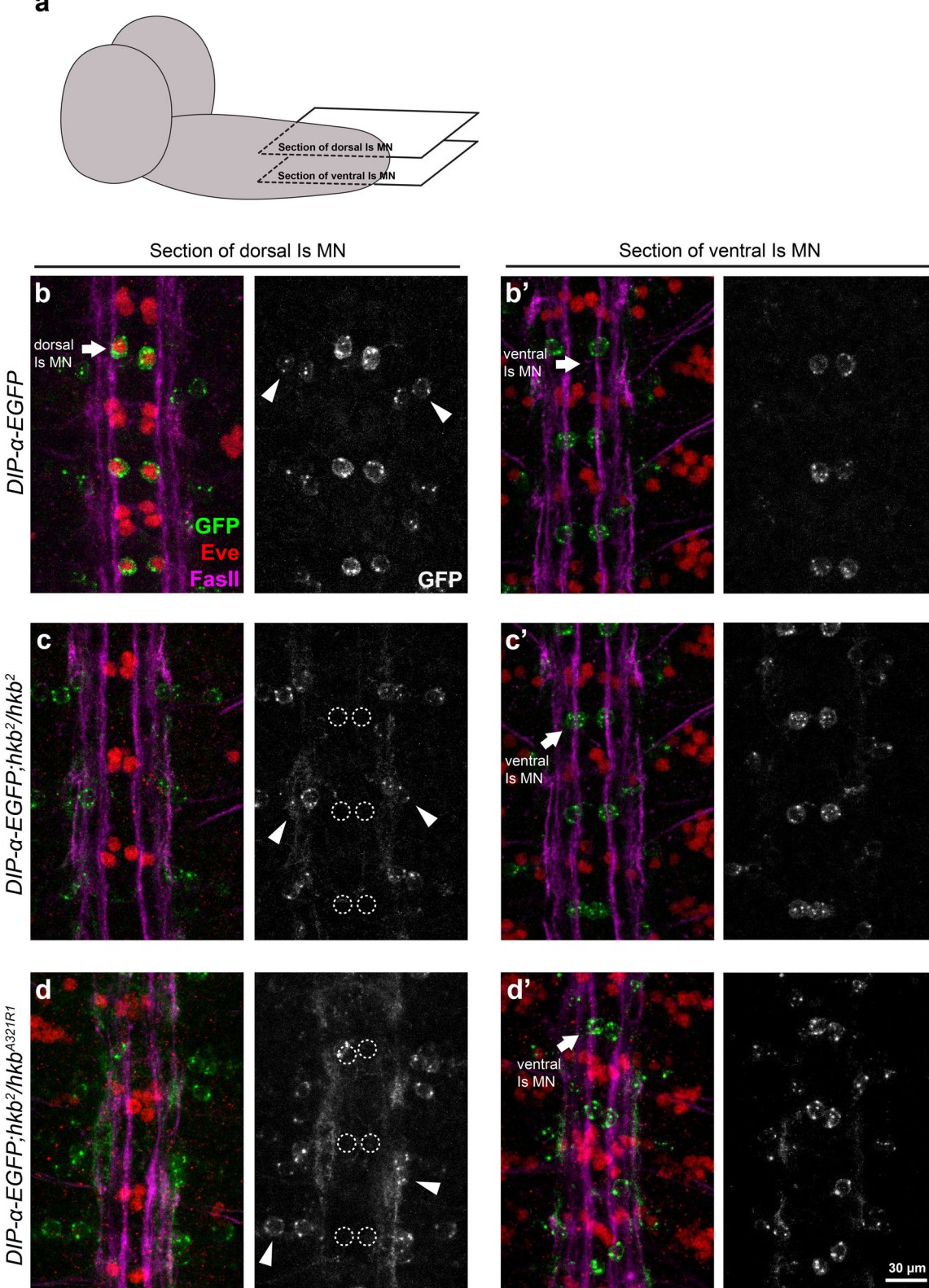

**Fig. 5 | *hkb* is required for *DIP-α-EGFP* expression in dorsal Is MNs. a** Cartoon depicting the focal planes in (**b–d**). Representative images of dorsal Is MN cell bodies (arrows in **b**) and ventral Is MN cell bodies (arrows in **b'**) labeled with GFP (green), Eve (red), and FasII (magenta) in control embryos. *DIP-α-EGFP* is expressed in both dorsal and ventral Is MNs. Note that there are also two interneurons on the dorsal side of each hemisegment that express DIP-α-EGFP (arrowheads in **b**). **c'** Representative images of dorsal Is MN cell bodies (dashed circles in **c**) and ventral Is MN cell bodies (arrows in **c'**) in *hkb²* mutant embryos. Eve and DIP-α-EGFP are missing in dorsal Is

MNs, whereas *DIP-α-EGFP* expression in ventral Is MNs is not affected. In addition, DIP-α-EGFP expressing interneurons are not affected (arrowheads in **c**), suggesting that Hkb function is specific to dorsal Is MNs. Representative images of dorsal Is MN cell bodies (dashed circles in **d**) and ventral Is MN cell bodies (arrows in **d'**) in heteroallelic *hkb* mutant embryos (*hkb²/hkb^{A321R1}*). Eve and DIP-α-EGFP are missing in dorsal Is MNs, whereas *DIP-α-EGFP* expression in ventral Is MNs is not affected. In addition, DIP-α-EGFP expressing interneurons are not affected (arrowheads in **d**), confirming that Hkb function is specific to dorsal Is MNs.

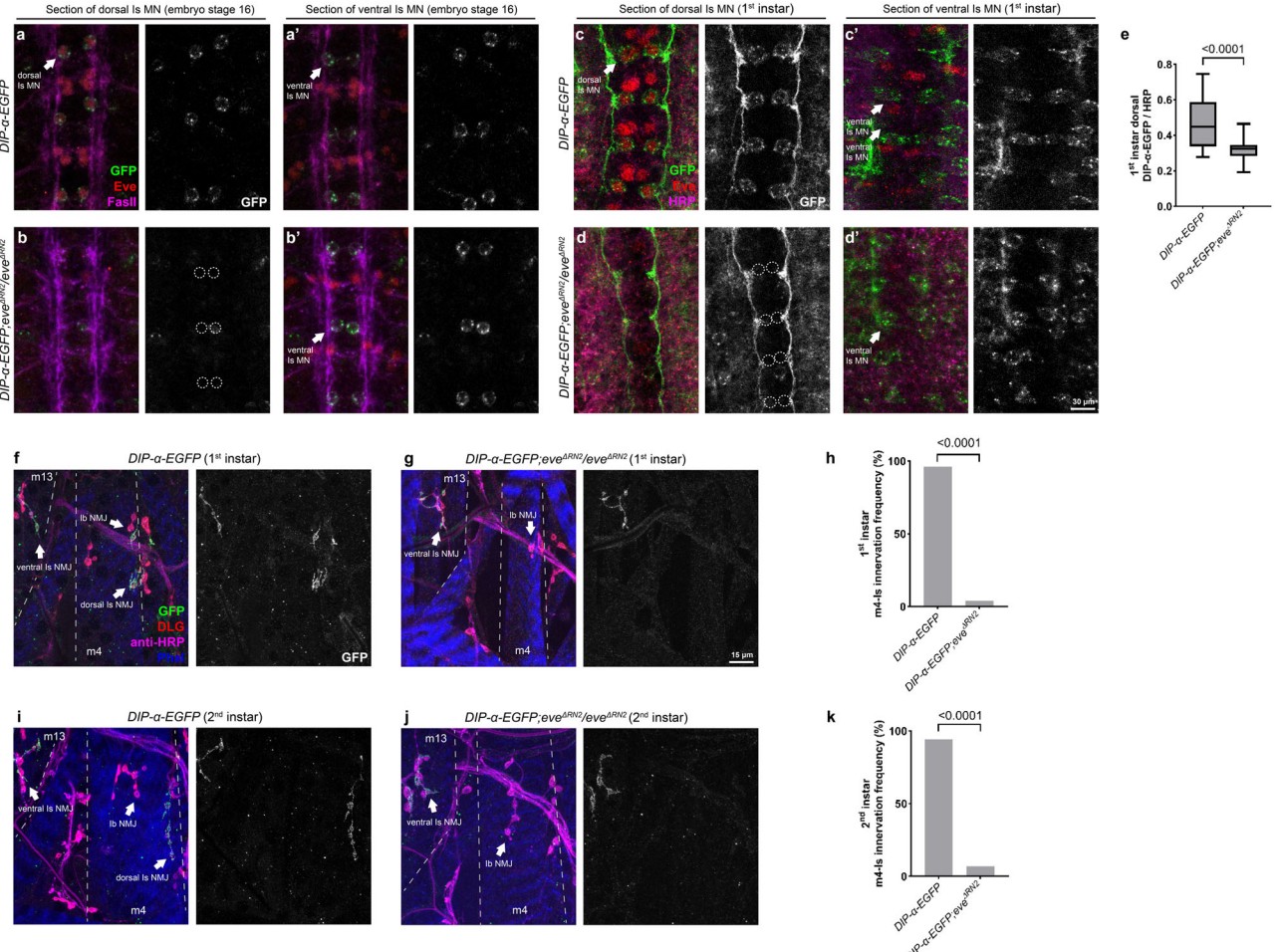

**Fig. 6 | *hkb* functions through *eve* to regulate *DIP-α-EGFP* expression and m4-Is innervation in embryos, 1st, and 2nd instar stages.** Representative images of dorsal Is MN cell bodies (arrows in **a**) and ventral Is MN cell bodies (arrows in **a'**) labeled with GFP (green), Eve (red), and FasII (magenta) in control embryos. *DIP-α-EGFP* is expressed in both dorsal and ventral Is MNs. Representative images of dorsal Is MN cell bodies (dashed circles in **b**) and ventral Is MN cell bodies (arrows in **b'**) in *eve^ARN2* mutant embryos. Eve and DIP-α-EGFP are missing in dorsal Is MNs, whereas *DIP-α-EGFP* expression in ventral Is MNs is not affected. Representative images of dorsal Is MN cell bodies (arrows in **c**) and ventral Is MN cell bodies (arrows in **c'**) labeled with GFP (green), Eve (red), and HRP (magenta) in control 1st instar larvae. *DIP-α-EGFP* is expressed in both dorsal and ventral Is MNs. Representative images of dorsal Is MN cell bodies (dashed circles in **d**) and ventral Is MN cell bodies (arrows in **d'**) in *eve^ARN2* mutant 1st instar larvae. Eve and DIP-α-EGFP are missing in dorsal Is MNs, whereas *DIP-α-EGFP* expression in ventral Is MNs is not affected. **e** Quantification of *DIP-α-EGFP* expression in dorsal Is MN cell bodies in control and *eve^ARN2* mutant larvae. N (ROI with two cell bodies) = 25 and 25. *p* value is indicated. Error bar indicates standard error of the mean (SEM). Representative images of NMJs formed by Ib MN, dorsal Is MN, and ventral Is MN (arrows) labeled with GFP (green), DLG (red), HRP (magenta), and Phalloidin (Blue), in 1st instar *DIP-α-EGFP* expressing larvae in (**f**) control and (**g**) *eve^ARN2* mutant background. **h** Quantification of m4-Is innervation frequency in 1st instar control and *eve^ARN2* mutant larvae. N (NMJs) = 27 and 25. *p* value is indicated. Representative images of NMJs formed by the Ib MN, dorsal Is MN, and ventral Is MN (arrows) labeled with GFP (green), DLG (red), HRP (magenta), and Phalloidin (Blue), in 2nd instar *DIP-α-EGFP* expressing larvae in (**i**) control and (**j**) *eve^ARN2* mutant background. **k** Quantification of m4-Is innervation frequency in 2nd instar control and *eve^ARN2* mutant larvae. N (NMJs) = 53 and 44. *p* value is indicated.

mechanisms regulate *DIP-α* expression in different MNs. Single-cell transcriptomics or candidate approaches in ventral Is MNs will aid in identifying other TFs that instruct *DIP-α* expression.

DIP-α is a member of the Dpr/DIP subfamilies of immunoglobulin CSPs. In a previous study, we mapped the expression of *dpr* and *DIP* genes in larval MNs and found that *dprs* were shared among many MNs and *DIPs* were more selectively expressed[10]. Interestingly, each of the 33 MNs expresses a unique subset of *dprs* and *DIPs* to reveal a cell-specific cell surface code. These data suggest that a highly complex regulatory transcriptional network is required to instruct the expression of these CSPs. An alternative but not mutually exclusive model is that one TF may regulate different CSPs in distinct neurons. A recent study in the fly olfactory circuit described a divergent "transcription factor to CSPs" relationship where the same TF, *acj6*, regulates many different CSPs in different cell types[32]. Further identification of the TF network in the larval nervous system will help to understand how the TF code is transmitted into a CSP code to guide synaptic recognition.

In summary, the regulatory programs controlling expression of circuit wiring molecules are more complex than we hypothesized. The plethora of recent single-cell RNA-seq data will undoubtedly shed light on candidate TFs, but follow-up genetic analyses will be required to confirm causal relationships between TFs and CSPs that underlie circuit wiring.

## Methods
### Genetics
The following *Drosophila* lines were used in this study: *w^111816*; *DIP-α-GAL4*[24]; *UAS-2×EGFP*; *dpr10^CR19*; *DIP-α^CR19*; *DIP-α-EGFP*[18]; *hkb^A321R1* (BL#2059)[35]; *hkb^2* (BL#5457)[55]; *eve^ARN243*. Df lines discussed in this paper are: ED5100 (BL#9226), ED5046 (BL#9197), ED5142 (BL#9198). All lines used for screen and sub-screen are listed in Supplementary Table 1[41,42]. For the

**Fig. 7 | *eve* genetically interacts with *DIP-α* and *hkb* to control of m4-Is innervation. a** Genetic interaction assay between *eve* and *DIP-α*. Single heterozygotes of *eve* or *DIP-α* did not have altered m4-Is innervation, while the trans-heterozygotes showed a further reduction, suggesting that *eve* genetically interacts with *DIP-α*. N (NMJs) = 102, 104, 84 and 104. *p* values are indicated. **b** Genetic interaction assay between *hkb* and *eve*. Single heterozygotes of *hkb* or *eve* did not have altered m4-Is innervation, while the trans-heterozygotes showed a further reduction, suggesting that *hkb* genetically interacts with *eve* to regulate Is innervation. N (NMJs) = 103, 87, 105, 81, 112 and 86. *p* values are indicated.

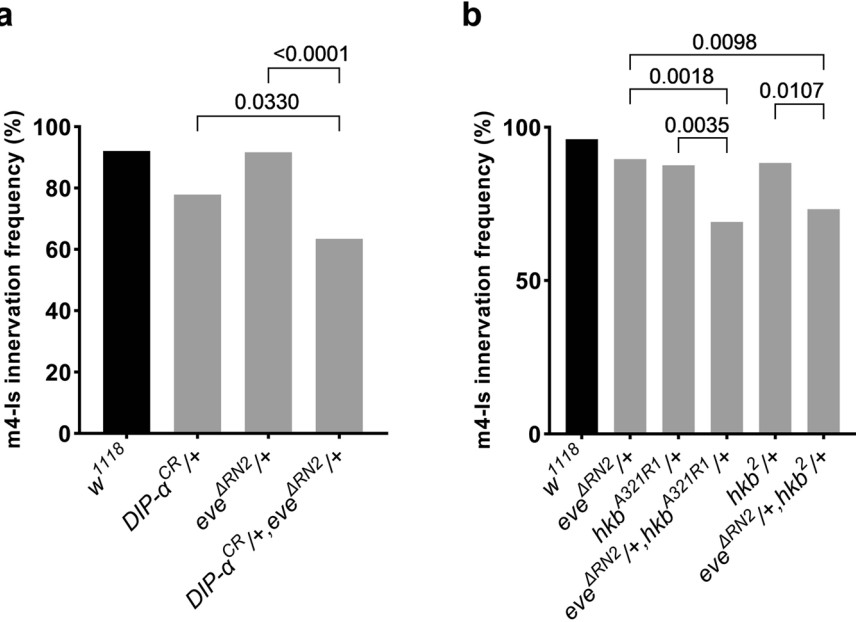

genetic screen, males from the Df lines or mutant lines were crossed to sensitized females (*DIP-α-GAL4; UAS-2×EGFP/CyO,actin > GFP; dpr10^CR^/ TM6,Tb,Hu*) and triple-heterozygous female larvae were selected to examine innervation frequency. For controls, *w^1118^* males were crossed to sensitized females to create trans-heterozygotes of *dpr10* and *DIP-α*. Screening workflow is illustrated in Fig. 1c (created with BioRender.com).

**Dissection, immunofluorescence, and imaging**
To examine Is MN innervation frequency, wandering first, second, and third instar larvae were dissected as previously described[56]. Briefly, larvae were collected and dissected on sylgard plates in PBS. Dissected fillets were fixed by 4% paraformaldehyde or Zamboni's solution for 20 min at room temperature and then washed three times in PBT (PBS with 0.05% Triton X-100). Samples were then blocked for 1 h in 5% goat serum (5% goat serum diluted in PBT) and incubated with primary antibodies at 4 °C overnight. Primary antibodies were then removed, and samples were washed for three times before 2 h incubation at room temperature (or overnight at 4 °C) with secondary antibodies. Finally, secondary antibodies were washed out and samples were washed and mounted in Vectashield (Vector Laboratories).

To examine *DIP-α* expression in embryos, stage 16 embryos were dissected as previous described[57]. Egg-laying chambers were set up with 40 females and 30 males and capped with grape juice plates. Eggs were collected for 2 h and grape juice plates covered in embryos were placed at 25 °C for 16 h for development. Embryos were staged under a Zeiss V20 stereoscope using the autofluorescence and morphology of the gut. Embryos at the correct stage were dechorionated and transferred onto a Superfrost Plus slides (Thermo Fisher Scientific, #22-037-246) covered by PBS. Embryos were then dissected with an electrolytically sharpened tungsten wire and stained similar to third instar samples. Antibodies used in this study were: rabbit anti-GFP (1:40k, gift from Michael Glozter, University of Chicago); rabbit anti-Eve (1:1000, gift from Ellie Heckscher, University of Chicago); mouse anti-Dlg (1:100, Developmental Studies Hybridoma Bank [DSHB] #4F3); mouse anti-FasII (1:100, DSHB #1D4); goat anti-rabbit Alexa 488 (1:500, Invitrogen #A11008); goat anti-rabbit Alexa 568 (1:500, Invitrogen #A11036); goat anti-mouse Alexa 568 (1:500, Invitrogen #A11031); goat anti-mouse Alexa 647 (1:500, Invitrogen #A32728); goat anti-HRP Alexa 405 (1:100, Jackson Immunological Research #123-475-021); goat anti-HRP Alexa 647 (1:100, Jackson Immunological Research #123-605-021).

Images were acquired on a Zeiss LSM800 confocal microscope using a 40X plan-neofluar 1.3 NA objective, or a 63X plan-apo 1.4 NA objective.

The same imaging parameters were applied to samples from the same set of experiments. Images were then analyzed and processed in ImageJ.

**Quantification of Is MN innervation frequency**
To examine Is MN innervation frequency, at least 6 third instar larvae were dissected and stained with anti-GFP (marker for Is MNs), anti-DLG and anti-HRP. Samples were visualized under a Zeiss AxioImager M2 scope with a Lumen light engine with a 20× Plan Apo 0.8 NA objective or an Olympus BX43 with an X-Cite 120LEDmini LED fluorescent illuminator. Is and Ib NMJs can be distinguished by bouton size, DLG intensity and whether it was GFP positive. If there was at least one Is bouton present on the muscle, it is scored as "innervated", otherwise it is scored as "not innervated". For each animal, m12, 13, 4 and 3 from abdominal hemisegments A2-A6 were assayed, and we collected a sample size of 50–60 hemisegments for each muscle. Innervation frequency was calculated as the percentage of "innervated" muscles in all muscles examined.

**Quantification of *DIP-α-EGFP* expression**
First instar larval brain pulls were conducted in PBS and mounted on poly-lysine coverslips. Intact brains were fixed and stained as described above. A Z-stack of the VNC was taken under 40X objective and a 2-slice projection covering the dorsal Is MN cell bodies was created in ImageJ (Sum slices). An area ROI was applied to the neuromere region spanning two MN cell bodies and EGFP and HRP intensity were measured. The same ROI size was used in this entire experiment. EGFP intensity was then normalized to HRP intensity to better reflect EGFP signal density.

**Statistics and reproducibility**
As we were mostly comparing the innervation frequency between two groups, we performed the Chi-square test followed by Yates' correction using Prism 8 software. Innervation frequencies and *p* values were reported in the figure legends. For the comparison of *DIP-α-EGFP* expression level in Fig. 6, unpaired *t* test with Welch's correction was performed (two-sided). Data were assumed to follow a Gaussian distribution.

**Reporting summary**
Further information on research design is available in the Nature Portfolio Reporting Summary linked to this article.

## Data availability
The source data behind the graphs in the paper can be found in Supplementary Data 1.

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

## Acknowledgements

This work is supported by NSF IOS-2048080, NINDS R01 NS123439 01, and a UChicago Faculty Diversity Grant to R.A.C., NINDS R15 NS101692 01 & 02 to C.G.V., and HHMI Gilliam GT16464 to R.J.S. This work is also supported by funds from the UChicago Biological Science Division, Committee of Developmental Biology and Department of Molecular Genetics & Cellular Biology, and by the Skidmore College Summer Collaborative Research Program. We thank the Bloomington Drosophila Stock Center (NIH P40OD018537) for fly lines. The monoclonal antibodies 4F3 and 1D4 were developed by Goodman, C. and they were obtained from the Developmental Studies Hybridoma Bank, created by the NICHD of the NIH and maintained at the University of Iowa, Department of Biology. We would like to thank Ellie Heckscher and Michael Glotzer for sharing resources. We thank the Skidmore Microscopy Imaging Center (SMIC) for the use of their microscopy resources. For their contributions to the screen, we also would like to thank the Spring 2018 Skidmore Neurophysiology students (Emily Blunt, Haoyang Huang, Jessy Idemoto, Dilhan Sirtalan, Julie Wang, Rob Warden, and Mary Beth Zahnleuter) and the Spring 2022 Skidmore Neurophysiology students (Jessica Auerbach, Margaret Besthoff, Gene Choi, Joa Comellas, DJ Flam, Melaina Gilbert, Kaylee Hua, Alexander Nardone, Bryan Taylor, and Victoria Thorpe). We would also like to thank Richard Fehon, David Pincus, Sunny Quinn, and members from the Carrillo laboratory for valuable discussions and comments.

## Author contributions

Y.W. and R.A.C designed research; Y.W., R.J.S., L.T.S., V.S., T.J.G., and B.W. performed experiments; Y.W. and R.J.S. analyzed data; Y.W. wrote the manuscript; R.J.S., C.G.V., and R.A.C. edited the manuscript.

## Competing interests

The authors declare no competing interests.
