## [Peer Review File · Communications Biology]

Reviewers' comments:

Reviewer #1 (Remarks to the Author):

This is a compact paper addressing the role of Hkb, Eve, and DIP-alpha in the dorsal Is motor neuron (henceforth dMN). The show that Hkb is required for expression of Eve in the dMN, and that Eve is required for expression of DIP-alpha in the dMN. The introduction is comprehensive (good!) and the manuscript is clearly written and easy to follow. The data are convincing and the figures nicely illustrate and support the main conclusions. I have several minor comments and no requested experiments. Very nice work.

- it is already known, and well cited here, that Hkb positively regulates Eve, and Eve regulates dMN target choice. Plus the Carrillo lab has shown that DIP-alpha is required for dMN target choice. Thus, it is unclear what novel findings are presented here. The authors should edit the abstract and discussion to highlight the open question they address, and their novel findings. Without these changes, most readers will wonder what is new here.

- in the abstract it says " huckebein (hkb), ... is important for target recognition specifically in the dorsal Is MN" however this is not data from this manuscript, but rather from a previous publication. Please adjust for accuracy.

- similarly, in the abstract it says " Genetic interaction between hkb and eve revealed that they act in the same pathway to regulate dorsal Is MN connectivity. " but only gene expression is characterized, previous work showed a role in connectivity. Please adjust for accuracy.

- introduction, page 5. Somewhere it should be cited that Bsh TF directly binds DIP-beta to control L4 lamina neuron synaptic specificity, published on BioRxiv (<https://doi.org/10.1101/2021.10.01.462699>).

Overall this is a beautiful manuscript that needs only minor text alterations.

Reviewer #2 (Remarks to the Author):

In this manuscript, Wang et al examine the regulation of cell surface protein required for target recognition using the Drosophila NMJ model system. Previous studies, including from Carrillo laboratory, demonstrated that DIP-alpha is exclusively expressed in two Is MNs. Also, the interaction between DIP-alpha and Dpr10 enables the recognition between the dorsal Is MN and some of its target muscles. The authors set up a sensitized background (trans-heterozygotes of DIP-alpha and Dpr10) to screen for other genes required for DIP-alpha&Dpr10-dependent target recognition. Here they report the results of screening through 105 Df lines covering the right arm of chromosome 3. The authors zoom onto one Df that showed enhanced innervation defects, narrowed down the region of interest, then tested eight candidate genes. They found that only huckebein (hkb), but not the other 7 genes evaluated, interact genetically with DIP-alpha & Dpr10. Interestingly, half of the individual genes tested showed significant rescue of the Is/m4 innervation phenotypes. Through one more round of genetic screening, the authors ruled out a hkb interaction with Dpr10 and confirmed the hkb-Dip-alpha interaction. Finally, they used an endogenously tagged DIP- α -EGFP allele to address the epistasis relationship between hkb and Dip-alpha and reported that hkb regulates Dip-alpha expression in an eve-dependent manner.

Overall, the manuscript is logically executed and beautifully written. But there are a few issues that must be resolved before publication.

Major issues:

1) This screen indicates that it's a lot easier to rescue than to worsen the m4 Is innervation defects (both with Df and individual genes). Yet the authors do not address the larger number of Dfs that restore the innervation, and instead focus only on the one case where they see a further reduction of m4 innervation.

I imagine that the authors are pursuing these different sets of modifiers in other manuscript(s). However, the results showed here must be acknowledged and discussed.

2) Figs 5 and 6 are weak and need additional arguments.

These two figures are not satisfying in the current format. Some staining is visible underneath the dotted circles in the dorsal Is panels in 4C-D and 5B. The dotted circles should be shown in only one panel (i.e. GFP only) so that the reader can appreciate the difference. Also, unrelated DIP-alpha-GFP-positive cells could be used as internal control to facilitate comparisons.

More importantly, the authors should perform confocal microscopy on (first or second instar) hkb and/or eve-deltaRN2 mutants:

- a) to document the innervation for Is dorsal and ventral (via HRP or Dlg staining)
- b) to compare Dip-alpha-EGFP expression levels in control and mutants

Other clarifications needed:

line 156- the sensitized background showed defects in Is innervation on m4 but not on m3. Why? Please describe and explain the lack of phenotypes in m3.

line 190- the candidate region contains tens of genes, yet the authors minimize this fact and only screen 8 genes. This needs to be detailed and clarified.

line 190 - the authors ignore lots of Dfs showing rescued innervation (see above) but also individual genes that show similar trends. Four genes show highly significant rescue in contrast to the mild reduction observed in the presence of the hkb[2] allele.

These are obvious and interesting results that must be dealt with.

Figure 4: The arguments will be a lot stronger if one more candidate(s) besides hkb is included for comparison.

Up till this point, most of the mutants (except hkb) show increased frequency of m4 innervation or no change when tested in the sensitized background. For Fig 4 the authors report a strong genetic interaction of hkb[2] with Dip-alpha, a lot stronger than with DIP-alpha&Dpr10 (Fig. 3). It will be reassuring to show that other candidates do not follow a similar trend.

225- include reference for Dip-alpha-EGFP KI

Figure 5: These results should be strengthened using hkb trans-allelic combinations that live until larval stages and showing Dip-alpha-EGFP and Is innervation in a hemisegment of the larval body wall muscles.

- document Is dorsal and ventral (via HRP or Dlg staining)
- compare Dip-alpha-EGFP expression levels

Figure 6: see comments for Figure 5 above -

These results promise to be even more exciting since eve may work as a suppressor.

274 and 294 "executor" may not be the best word.

February 20, 2024

Dear Reviewers,

Thank you for the constructive feedback and insightful comments and suggestions. To address each of your comments, we include a detailed point-by-point response below.

Thank you again and we look forward to hearing back from you.

Sincerely,

Robert Carrillo

Reviewers' comments:

Reviewer #1 (Remarks to the Author):

This is a compact paper addressing the role of *Hkb*, *Eve*, and *DIP-alpha* in the dorsal l5 motor neuron (henceforth dMN). The show that *Hkb* is required for expression of *Eve* in the dMN, and that *Eve* is required for expression of *DIP-alpha* in the dMN. The introduction is comprehensive (good!) and the manuscript is clearly written and easy to follow. The data are convincing and the figures nicely illustrate and support the main conclusions. I have several minor comments and no requested experiments. Very nice work.

Thank you!

- it is already known, and well cited here, that *Hkb* positively regulates *Eve*, and *Eve* regulates dMN target choice. Plus the Carrillo lab has shown that *DIP-alpha* is required for dMN target choice. Thus, it is unclear what novel findings are presented here. The authors should edit the abstract and discussion to highlight the open question they address, and their novel findings. Without these changes, most readers will wonder what is new here.

Reply: We thank this reviewer for their suggestions and agree that the regulation and role of *hkb* and *eve* were known before. However, the downstream CSP(s) that controls target recognition was not known. Our study finds that *DIP-α* is one of these downstream CSPs and reveals part of the regulatory mechanism upstream *DIP-α*. We reinforced these novelties in this revised

manuscript. Please see revised text: Abstract Line 45-48, Introduction Line 119-120 and Line 136-139, Discussion Line 273-280).

- in the abstract it says " huckebein (*hkb*), ... is important for target recognition specifically in the dorsal Is MN" however this is not data from this manuscript, but rather from a previous publication. Please adjust for accuracy.

Reply: We apologize for this inaccuracy. We updated the phrase in abstract Line 51-53.

- similarly, in the abstract it says " Genetic interaction between *hkb* and *eve* revealed that they act in the same pathway to regulate dorsal Is MN connectivity. " but only gene expression is characterized, previous work showed a role in connectivity. Please adjust for accuracy.

Reply: We agree with this reviewer that a previous study showed the role of *hkb* and *eve* in connectivity in *hkb* and *eve* mutants. However, our genetic interaction tests (current Figure 7B) between *hkb* and *eve* also confirmed their role, and more importantly, suggested that they act in the same pathway for m4-Is recognition.

- introduction, page 5. Somewhere it should be cited that Bsh TF directly binds DIP-beta to control L4 lamina neuron synaptic specificity, published on BioRxiv (<https://doi.org/10.1101/2021.10.01.462699>).

Reply: We thank this reviewer for bringing up this very important reference that we neglected to include! We discussed their findings and cited this paper in Line 106-108.

Overall this is a beautiful manuscript that needs only minor text alterations.

Thank you!

Reviewer #2 (Remarks to the Author):

In this manuscript, Wang et al examine the regulation of cell surface protein required for target recognition using the *Drosophila* NMJ model system. Previous studies, including from Carrillo laboratory, demonstrated that DIP-alpha is exclusively expressed in two Is MNs. Also, the interaction between DIP-alpha and Dpr10 enables the recognition between the dorsal Is MN and some of its target muscles. The authors set up a sensitized background (trans-heterozygotes of DIP-alpha and Dpr10) to screen for other genes required for DIP-alpha&Dpr10-dependent target recognition. Here they report the results of screening through 105 Df lines covering the right arm of chromosome 3. The authors zoom onto one Df that showed enhanced innervation defects, narrowed down the region of interest, then tested eight candidate genes. They found that only huckebein (*hkb*), but not the other 7 genes evaluated, interact genetically with DIP-alpha & Dpr10. Interestingly, half of the individual genes tested showed significant rescue of the Is/m4 innervation phenotypes. Through one more round of genetic screening, the authors ruled out a *hkb* interaction with Dpr10 and confirmed the *hkb*-Dip-alpha interaction. Finally, they used an endogenously tagged DIP- α -EGFP allele to address the epistasis relationship between *hkb* and Dip-alpha and reported that *hkb* regulates Dip-alpha expression in an *eve*-dependent manner.

Overall, the manuscript is logically executed and beautifully written. But there are a few issues that must be resolved before publication.

Thank you!

Major issues:

1) This screen indicates that it's a lot easier to rescue than to worsen the m4 Is innervation defects (both with Df and individual genes). Yet the authors do not address the larger number of Dfs that restore the innervation, and instead focus only on the one case where they see a further reduction of m4 innervation.

I imagine that the authors are pursuing these different sets of modifiers in other manuscript(s). However, the results showed here must be acknowledged and discussed.

Reply: We appreciate this reviewer highlighting that most Dfs and individual genes rescue the innervation. Indeed, we were surprised to see that most lines restored innervation. We are following up some of these candidates, but their roles are not yet determined.

One hypothesis for this phenomenon is that there are numerous repulsive pathways that contribute to synaptic connectivity in the neuromuscular circuit. The partial loss of components in these pathways combined with the transheterozygous *DIP-α/+; dpr10/+* background would lead to rescue of m4 innervation by the Is motor neuron. However, the candidate genes may act independently of *DIP-α* and *Dpr10*. Additionally, the mechanisms driving m4 connectivity are likely very limited based on the fact that *DIP-α* or *dpr10* null mutants lead to complete loss of m4-Is innervation. Thus, candidate genes that exacerbate the innervation phenotype in the sensitized background would include genes that alter expression and localization of *DIP-α* or *dpr10* and potential signaling molecules in the *DIP-α-Dpr10* signaling pathway.

We updated the Discussion to include these possibilities (Line 284-296).

2) Figs 5 and 6 are weak and need additional arguments.

These two figures are not satisfying in the current format. Some staining is visible underneath the dotted circles in the dorsal Is panels in 4C-D and 5B. The dotted circles should be shown in only one panel (i.e. GFP only) so that the reader can appreciate the difference.

Reply: We removed the dotted circles in the colored panel as this reviewer suggested. Please see current Figure 5C, 5D, 6B and 6D.

Also, unrelated *DIP-α-GFP*- positive cells could be used as internal control to facilitate comparisons.

Reply: We thank the reviewer for this suggestion. Indeed, we must exclude the possibility that *Hkb* affects *DIP-α* expression globally. We updated our representative images and now show two *DIP-α-EGFP* positive interneurons (arrowheads in Figure 5B) that are not affected by *hkb* mutations (arrowheads in Figure 5C and 5D). These cells serve as a control confirming *Hkb* function is specific to the dorsal Is MNs. We updated the figure legend and text Line 238-239.

More importantly, the authors should perform confocal microscopy on (first or second instar) *hkb* and/or *eve-deltaRN2* mutants:

Reply: We agree with this reviewer that representative images of the NMJ and quantification of *DIP-α-EGFP* expression will be helpful. Because *hkb* mutants do not survive to later embryo stages when NMJ formation occurs, we examined 1st and 2nd instar *eve^{ΔRN2}* mutants to address these requests (Line 258-261).

a) to document the innervation for Is dorsal and ventral (via HRP or Dlg staining)

Reply: The innervation of ventral Is MN is not affected, but the innervation of dorsal Is MN is reduced. We include representative NMJ images and the quantification of innervation frequency in current Figure 6F-6K.

b) to compare Dip-alpha-EGFP expression levels in control and mutants

Reply: The DIP- α -EGFP expression level in the VNC is significantly reduced in *eve* ^{Δ RN2} mutants. We included a representative image of a 1st instar VNC and quantification in current Figure 6C-6E.

Other clarifications needed:

line 156- the sensitized background showed defects in Is innervation on m4 but not on m3. Why? Please describe and explain the lack of phenotypes in m3.

Reply: DIP- α -Dpr10 interaction is 100% necessary for m4-Is innervation, but is not 100% required for m3-Is innervation (Ashley et al., 2019). This could possibly be due to other redundant CSPs that guide m3 and Is recognition. Therefore, we do not anticipate a change of m3-Is innervation frequency in this sensitized background, as it is only a trans-heterozygote of *DIP- α* and *dpr10*.

line 190- the candidate region contains tens of genes, yet the authors minimize this fact and only screen 8 genes. This needs to be detailed and clarified.

Reply: We apologize that we did not make it clear. The reason that we chose these candidates are based on their putative roles in neural development and the availability of verified mutant stocks. This candidate region (4197kb to 4453kb) covers 58 genes including protein coding genes and non-coding RNAs. Eight of them (examined in this study) likely functioning in nervous system and have corresponding stocks available. We added these criteria in the current manuscript (Line 192-194).

line 190 - the authors ignore lots of Dfs showing rescued innervation (see above) but also individual genes that show similar trends. Four genes show highly significant rescue in contrast to the mild reduction observed in the presence of the *hkb*[2] allele. These are obvious and interesting results that must be dealt with.

Reply: We appreciate this reviewer for highlighting these candidates. The reason that we decided to first follow up *hkb* even though it shows a mild reduction is due to its role as a transcription factor and its expression in the dorsal Is MN. Additionally, we were interested in regulatory programs that instruct expression of connectivity genes.

We also looked into the candidates that rescue the innervation phenotype, but given their proposed cellular function, it is difficult to speculate how they are involved in the DIP- α -Dpr10 pathway. As described above they likely act independent of DIP- α and Dpr10. For example, we observed larger NMJs (more boutons) and elevated synaptic activity in *cpx* mutants, which is consistent with previous studies (Banerjee et al., 2021; Choi et al., 2014). Thus, the increase of innervation frequency in triple-heterozygotes (*DIP- α /+*, *dpr10/+*, *cpx/+*) could be due to the

elevated synaptic activity which has been shown to perturb synaptic pruning. We updated the manuscript to highlight these possibilities in the Discussion section Line 284-296.

Figure 4: The arguments will be a lot stronger if one more candidate(s) besides *hkb* is included for comparison.

Up till this point, most of the mutants (except *hkb*) show increased frequency of m4 innervation or no change when tested in the sensitized background. For Fig 4 the authors report a strong genetic interaction of *hkb*[2] with *Dip-alpha*, a lot stronger than with *DIP-alpha&Dpr10* (Fig. 3). It will be reassuring to show that other candidates do not follow a similar trend.

Reply: The genetic interaction between *hkb*² and *DIP-α* in Figure 4 is actually the same penetrance as its interaction with the *DIP-α/+; dpr10/+* transheterozygote in Figure 3; they are both approximately 40%, suggesting that *hkb*² is only interacting with *DIP-α*. The difference in significance level (* in Figure 3 vs <0.0001 in Figure 4) is due to the control in each experiment – in Figure 3 the control is the sensitized background (50%), whereas in Figure 4 the control is *DIP-α/+* (~70%).

The reviewer brings up an interesting point to test additional candidate genes to see how they interact with *DIP-α* or *dpr10*. Notably, the sensitized background (*DIP-α/+; dpr10/+*) enables identification of *Dfs*/genes that “worsen” or “rescue” the m4-Is innervation frequency. However, single heterozygotes of *DIP-α* and *dpr10* impede discovery of candidates that rescue because the m4-Is innervation frequency in single heterozygotes of *DIP-α* and *dpr10* is near wild type level. Therefore, we would not be able to determine genetic interactions between *DIP-α* and candidate genes that rescue. In addition, as discussed above, we examined single heterozygotes of a subset of these candidates, and they show m4-Is innervation frequency at wild type levels, suggesting that they may act independently of *DIP-α* and *dpr10*.

225- include reference for *Dip-alpha*-EGFP KI

Reply: We thank the reviewer and have added the missing reference in Line 228.

Figure 5: These results should be strengthened using *hkb* trans-allelic combinations that live until larval stages and showing *Dip-alpha*-EGFP and Is innervation in a hemisegment of the larval body wall muscles.

- document Is dorsal and ventral (via HRP or Dlg staining)
- compare *Dip-alpha*-EGFP expression levels

Reply: We appreciate the reviewer’s suggestion. Unfortunately, *hkb*² and *hkb*^{A321R1} are both null alleles. Single mutant or trans-allelic *hkb* embryos do not develop to the larval stage. We tried to create a conditional *hkb* knockout but failed due to its close proximity to FRT82B and the centromere. In addition, we tested *hkb* RNAi lines (*TRiP.JF03410* and *TriP.HMS01216*), but the larvae of *actin>GAL4; UAS-hkb-RNAi* are completely wild type, suggesting knockdown was very mild or none at all.

To address this request and the request below, we examined these questions in *eve*^{ΔRN2} larvae. Please see next reply.

Figure 6: see comments for Figure 5 above -

These results promise to be even more exciting since *eve* may work as a suppressor.

Reply: We performed the recommended experiment and provided results from 1st and 2nd instar *eve*^{ΔRN2} mutants. We observed no m4-Is innervation and no *DIP-α-EGFP* expression in dorsal Is MNs (Figure 6C-6K). In addition, we agree with this reviewer that it is of great interests to examine how *Eve* controls *DIP-α* expression since *Eve* is generally considered a repressor. However, this is beyond the scope of this paper and will be followed-up in future work.

274 and 294 "executor" may not be the best word.

Reply: We deleted "executor" and instead, used "molecule".

Ashley, J., Sorrentino, V., Lobb-Rabe, M., Nagarkar-Jaiswal, S., Tan, L., Xu, S., Xiao, Q., Zinn, K. and Carrillo, R. A. (2019). Transsynaptic interactions between IgSF proteins *DIP-α* and *Dpr10* are required for motor neuron targeting specificity. *eLife* 8, e42690.

Banerjee, S., Vernon, S., Jiao, W., Choi, B. J., Ruchti, E., Asadzadeh, J., Burri, O., Stowers, R. S. and McCabe, B. D. (2021). Miniature neurotransmission is required to maintain *Drosophila* synaptic structures during ageing. *Nat Commun* 12, 4399.

Choi, B. J., Imlach, W. L., Jiao, W., Wolfram, V., Wu, Y., Grbic, M., Cela, C., Baines, R. A., Nitabach, M. N. and McCabe, B. D. (2014). Miniature Neurotransmission Regulates *Drosophila* Synaptic Structural Maturation. *Neuron* 82, 618–634.

REVIEWERS' COMMENTS:

Reviewer #1 (Remarks to the Author):

All of my comments have been satisfactorily addressed. It is a nice paper that is ready to publish.

Reviewer #2 (Remarks to the Author):

In the revised manuscript the authors have addressed all my comments and concerns.

Congratulations on a work beautifully done!